# Niche Differentiation of Active Methane-Oxidizing Bacteria in Estuarine Mangrove Forest Soils in Taiwan

**DOI:** 10.3390/microorganisms8081248

**Published:** 2020-08-17

**Authors:** Yo-Jin Shiau, Chiao-Wen Lin, Yuanfeng Cai, Zhongjun Jia, Yu-Te Lin, Chih-Yu Chiu

**Affiliations:** 1Department of Bioenvironmental Systems Engineering, National Taiwan University, Taipei 106, Taiwan; yshiau@ncsu.edu; 2Biodiversity Research Center, Academia Sinica, Nangang, Taipei 11529, Taiwan; clin10@ncsu.edu (C.-W.L.); d87241004@ntu.edu.tw (Y.-T.L.); 3State Key Laboratory of Soil and Sustainable Agriculture, Institute of Soil Science, Chinese Academy of Sciences, Nanjing 210008, China; yfcai@issas.ac.cn

**Keywords:** aerobic methane oxidation, methanotrophs, coastal mangrove soil, DNA stable isotope probing, *pmoA* gene, 16S rRNA gene

## Abstract

Mangrove forests are one of the important ecosystems in tropical coasts because of their high primary production, which they sustain by sequestering a substantial amount of CO_2_ into plant biomass. These forests often experience various levels of inundation and play an important role in CH_4_ emissions, but the taxonomy of methanotrophs in these systems remains poorly understood. In this study, DNA-based stable isotope probing showed significant niche differentiation in active aerobic methanotrophs in response to niche differentiation in upstream and downstream mangrove soils of the Tamsui estuary in northwestern Taiwan, in which salinity levels differ between winter and summer. *Methylobacter* and *Methylomicrobium*-like Type I methanotrophs dominated methane-oxidizing communities in the field conditions and were significantly ^13^C-labeled in both upstream and downstream sites, while *Methylobacter* were well adapted to high salinity and low temperature. The Type II methanotroph *Methylocystis* comprised only 10–15% of all the methane oxidizers in the upstream site but less than 5% at the downstream site under field conditions. ^13^C-DNA levels in *Methylocystis* were significantly lower than those in Type I methanotrophs, while phylogenetic analysis further revealed the presence of novel methane oxidizers that are phylogenetically distantly related to Type Ia in fresh and incubated soils at a downstream site. These results suggest that Type I methanotrophs display niche differentiation associated with environmental differences between upstream and downstream mangrove soils.

## 1. Introduction

Global warming is caused by anthropogenic greenhouse gas emissions, and the major greenhouse gas, methane (CH_4_), traps 25 times more heat than carbon dioxide (CO_2_) over a 100-year time horizon [1]. Globally, about 40% of CH_4_ is released from natural ecosystems, mainly anaerobic environments such as rice paddies and wetlands [2,3]. However, the global CH_4_ budget varies greatly, which is in part because biogeochemical models do not include microbially-mediated processes. For instance, it has been widely accepted that 30–90% of CH_4_ produced in anaerobic ecosystems is consumed by aerobic methane-oxidizing bacteria (methanotrophs) when it diffuses through shallow soils [4]. Linking the taxonomic identity of aerobic methanotrophs to the ecologically important process of CH_4_ oxidation is key to developing a mechanistic model that accurately assesses the global CH_4_ emission flux from wetlands.

Indeed, methanotrophic bacteria are found throughout anaerobic ecosystems such as rice paddies, lakes, and geothermal springs [5,6,7,8], but their compositions may vary depending on their environmental niches [9,10]. Methanotrophic communities can generally be categorized into three types: Type I, which includes the family Methylococcaceae (γ-Proteobacteria) [11]; Type II, which includes the families Methylocystaceae and Beijerinckiaceae (α-Proteobacteria); and Type X, which includes some other γ-Proteobacteria [12,13]. Almost all of these above mentioned methanotrophs possess the particulate methane monooxygenase (pMMO), which catalyzes CH_4_ oxidation [14,15]. By using functional gene biomarkers (i.e., *pmoA*), previous studies found that Type II methanotrophs often dominate CH_4_ oxidation in freshwater ecosystems such as rice paddies, while Type I methanotrophs are mostly found in saline water ecosystems such as hypersaline lakes, estuaries, and coastal wetlands [16,17,18,19,20,21]. Recent studies further revealed niche differentiation in Type I methanotrophs at a finer resolution, demonstrating that Type Ia methanotrophs have a high tolerance to salinity (i.e., >1% NaCl), while Type Ib methanotrophs do not [22,23].

It has long been assumed that Type I methanotrophs grow in and respond quickly to eutrophic conditions but decrease in abundance rapidly in nutrient-limited conditions or barren environments [9,10,21]. Type I methanotrophs are often considered r-selected species because they have high reproduction rates but low survival rates under adverse environments. On the other hand, Type II methanotrophs are adapted to oligotrophic and fluctuating environments but have low growth rates under changing environments, and are thus considered K-selected organisms (i.e., low reproduction rates but high survival rates) [9].

DNA-based stable isotope probing (DNA-SIP) is a powerful means to establish a direct link between CH_4_ oxidation and taxonomic identity for active methanotrophs in complex environments [4], and the combination of DNA-SIP and high-throughput sequencing of *pmoA* and 16S ribosomal RNA (16S rRNA) genes further enables the identification of active methanotrophs at an unprecedented resolution [19,20,24]. However, many studies have only focused on freshwater ecosystems, especially rice fields. A recent study showed that Type I methanotrophs such as *Methylosarcina*, *Methylobacter,* and *Methylomonas* made up the most abundant and active CH_4_-oxidizing microbial community in coastal mangrove soils, and the compositions of these genera varied with different soil aeration statuses [19]. However, it remains poorly understood how active methanotrophs respond to environmental divergences in coastal ecosystems.

In this study, we evaluated the variations in methanotrophic community compositions in two mangrove forests along the Tamsui estuary in Taiwan, which covers salinity ranges from mesohaline (5–18 psu) to polyhaline (18–30 psu) [25], and compared the results to our previous work, which was performed in a space between these two forests [19]. This study applied DNA-SIP, with the objective of determining whether there is a correlation between CH_4_ oxidation kinetics and biogeographic patterns of methanotrophs in mangrove forests along a salinity gradient.

## 2. Materials and Methods

### 2.1. Site Description and Soil Sampling

The two mangrove forests sampled in this study are located in the Tamsui estuary, Taipei, Taiwan (Figure 1). The climate is mild and dry in winter and hot and wet in summer. The mean atmospheric temperature in the Tamsui River is 18.6 and 31.5 °C in winter and summer, respectively, and the average water temperature is 18.3 and 30.5 °C, respectively. The downstream sampling site, Bali, is located 1 km from the mouth of the Tamsui River, and the upstream sampling site, Guandu, is located 8 km from the estuary. *Kandelia obovate* is the dominant vegetation, and *Phragmites communis* and *Cyperus malaccensis* are scattered along the edges of both mangrove forests [25,26]. Because of its geographical locations, the average salinity of Tamsui River at the time of the study was 21 psu downstream and 17 psu upstream.

Soil samples were collected in January and August of 2018 to estimate the dynamics of methanotroph communities across seasons. During each field sampling, 25 soil core samples were collected from each site with polyvinyl chloride (PVC) tubes 1.5 cm in diameter and 30 cm long. Core samples were collected about every 5–10 m on a random walk, covering about 0.7 and 0.5 ha of the upstream and downstream mangrove forests, respectively. The soil samples were stored in ice and transported to the laboratory for analysis. After the collected soils were retrieved from the tubes, the top 2 cm of the bulk soil samples were sliced off using a knife pretreated with 70% ethanol. Any visible litter/roots were manually removed from the samples. Then, the 2-cm soil samples from the 25 cores from each site were mixed carefully to form a composite sample and stored in a plastic sealed bag at 4 °C for further analyses.

The soil texture was determined by a pipette method [27]. A 1:1 soil to water slurry was created and measured by a pH meter with a glass electrode (Jenco 6009, Jenco Instruments, San Diego, CA, USA). Soil total organic C (TOC) and total N (TN) were evaluated by the combustion method with an elemental analyzer (Fisons NA1500, ThermoQuest Italia, Milan, Italy). Soil labile C and N (soluble organic C: S_b_OC; soluble organic N: S_b_ON; ammonium: NH_4_^+^; nitrate: NO_3_^−^; total dissolved N: TDN) were extracted using a 2 M KCl extraction method [28,29]. Briefly, extracts were filtered from 5 g of soil and soaked with 50 mL KCl for an hour at 150 rpm. Then, the extracts were analyzed with the cadmium reduction method for NO_3_^−^ [30], indophenol method for NH_4_^+^ [31], and persulfate method for TDN [32]. S_b_OC was analyzed with a TOC analyzer (1010, Analytical, TX, USA). Soil potential mineralizable N (PMN) was measured using a waterlogging incubation method [33].

### 2.2. Aerobic CH_4_ Oxidation Experiment and DNA-SIP Gradient Fractionation

Six replicates of soil samples from each site in January and August were incubated under oxic conditions. For each replicate, 10 g of soil was placed into a 125-mL serum bottle, then the bottle was sealed with a rubber stopper and covered tightly with an aluminum cap. Three replicates were injected with 3 mL of pure (99 atom% ^13^C) ^13^CH_4_ (Sigma–Aldrich, St. Louis, MO, USA) as the isotope-labeled set, and the other three replicates were injected with the same amount (99 atom% ^12^C) of ^12^CH_4_ to create a headspace of about 1.8% CH_4_. Then, the samples were placed in a dark incubator at 20 °C for 7 days to mimic the average soil temperature in the field (LM-570RD, Yihder Technology Co., Taiwan).

To measure the CH_4_ concentrations, a syringe was used to extract a 0.5-mL gas sample from the headspace of each bottle. The CH_4_ concentration of the sample was immediately measured by gas chromatography with a flame ionization detector (GC-FID, GC9720, Fuli Instruments, Zhejiang, China). All the CH_4_ had nearly depleted by day 7, so the CH_4_ uptake rate was determined based on two data points between Day 0 (initial) and Day 7.

After the incubation experiment was completed, about 0.8 g of incubated soil from each replicate was extracted and its total genomic DNA was isolated with the PowerSoil^®^ DNA isolation kit (MO BIO Laboratories, Inc., Carlsbad, CA, USA) based on the manufacturer’s instructions. DNA was also extracted from the fresh soils under field conditions. All the extracted DNA was stored at −20 °C until sequence analysis. Subsamples of the total genomic DNA were used to evaluate the absolute abundance of *pmoA* genes by real-time quantitative polymerase chain reaction (qPCR) analysis with a primer pair (A189F/mb661r) [34] and qPCR reagent kit (RR420A, SYBR^®^ Premix Ex Taq™, Takara Bio Inc., Japan). The samples were amplified using the following cycling steps: 5 min at 95 °C, followed by 40 cycles at 92 °C for 10 s, 55 °C for 30 s, 72 °C for 30 s, and 80 °C for 12 s with acquisitions. Analysis of the *pmoA* copies yielded 98% efficiency and R^2^ = 0.9997.

The total DNA extracted from ^13^CH_4_ and ^12^CH_4_ incubated samples was separated into 15 density gradient fractions with an ultracentrifuge (Optima XPN-80, Beckman Coulter, Brea, CA, USA) with isopycnic density gradient centrifugation using the fractionation technique summarized in Lu and Jia [35]. Then, a digital refractometer (AR200, Reichert Tech., Ametek Inc., NY, USA) was used to measure the buoyant density for each DNA fraction. The same qPCR procedure was applied to evaluate the abundance of *pmoA* genes in fractions 2–14. The active methanotroph communities under high CH_4_ concentrations were identified by comparing the abundance of *pmoA* genes in fractions 2–14 between the ^13^CH_4_ and ^12^CH_4_ samples.

### 2.3. Analyses of Active Methanotrophic Communities

The fraction of ^13^CH_4_ samples with the highest abundance of *pmoA* genes was analyzed by high-throughput sequencing to evaluate active methanotroph compositions. The methanotrophic species from fresh soils were also investigated using the total extracted DNA. The polymerase chain reaction (PCR) technique was used to amplify *pmoA* gene fragments with the primers A189F/mb661r and reagent kit RR902A, Premix Ex Taq™ (Takara Bio Inc., Kusatsu, Shiga, Japan) [16]. The PCR amplicons were sequenced on an Illumina MiSeq instrument with a MiSeq Reagent Kit v3 (2 × 300 bp paired-end reads).

Mothur was applied to process and classify the sequences based on the instructions and reference database in Dumont et al. (2014). The FunGene Pipeline [36] was used with USEARCH 6.0 Chimera Check [37] and Expand Mappings Programs to remove chimeric reads and obtain representative operational taxonomic units (OTUs), respectively. The microbial community diversity was calculated with the Shannon diversity index through the FunGene Pipeline.

To evaluate the total bacterial community in soil under field and incubated conditions, a universal primer pair (515F/907R) and reagent kit (KAPA HiFi HotStart ReadyMix, KAPA Biosystems, MA, USA) were used to amplify the V4 region of the 16S rRNA gene; the total bacterial composition was also analyzed by the Illumina Miseq platform. The sequences were analyzed by the Ribosomal Database Project (RDP) pipeline (http://pyro.cme.msu.edu/) and the microbial communities were classified by the naïve Bayesian rRNA classifier [38] to determine the soil microbial compositions. The microbial communities associated with methanotrophs were selected for analysis.

### 2.4. Statistical Analyses

The differences in soil physiochemical properties, CH_4_ oxidation potential, and *pmoA* gene abundance between the two studied mangrove sites were tested with a one-way analysis of variance (one-way ANOVA) and Tukey’s honestly significant difference (HSD) with JMP 11.0 (SAS Inc., Cary, NC, USA). *p*-value < 0.05 was considered significant. Based on the suggestions from Shiau et al. [20], the qPCR copies of *pmoA* were log transformed before performing any statistical analyses to normalize the data.

To evaluate the relationships between soil physiochemical properties and in situ methanotrophic communities, a canonical correspondence analysis (CCA) was applied to analyze the effects of environmental factors on methanotroph composition with XLSTAT 2019 (Addinsoft Inc., New York, NY, USA). Phylogenetic trees of methanotrophic communities in the studied mangrove forests were analyzed using MEGA X (Pennsylvania State University, State College, PA, USA). The representative sequences can be accessed on the National Taiwan University Scholars website (https://scholars.lib.ntu.edu.tw/handle/123456789/510011) and in the Appendix A.

## 3. Results

The total soil TOC and TN were not significantly different between the two seasons (Table 1). For the KCl extracts, the S_b_OC appeared to be higher in winter than summer (*p* < 0.05) and the upstream site than the downstream site. In addition, soil soluble N—i.e., NH_4_^+^ (*p* < 0.001), TDN (*p* < 0.02), and PMN (*p* < 0.05)—were higher downstream in summer, while labile N was similar at both sites in winter (*p* > 0.05).

The CH_4_ oxidation potentials were similar between mangrove soils, except in samples from the downstream site in winter (Figure 2). This difference in CH_4_ oxidation potentials at the downstream site in winter was also reflected in their methanotrophic bacterial communities. qPCR results showed that the *pmoA* copy numbers in the fresh mangrove soils were highest at the downstream site in winter. In addition, *pmoA* copy number in the incubated soils increased by approximately one order of magnitude in both mangrove soils for both seasons (Figure 3).

Gradient fractionation of the ^13^CH_4_-incubated soil DNA showed that the ^13^C-labeled *pmoA* genes were mostly concentrated in the 7–8 fractions (density: 1.733–1.737 g mL^−1^), except in the soil collected upstream in summer, in which the heavy fraction of the *pmoA* genes was located in the 10th fraction (density: 1.724 g mL^−1^) (Figure 4). In addition, regional peak fractions (i.e., 10th fractions in Figure 4c,d) of the *pmoA* gene were also found in the Bali mangrove soils.

Results of high-throughput sequencing of the *pmoA* genes revealed that Type Ia *Methylobacter* was the most dominant methanotrophic population, making up 55–58% and 54–56% of bacterial methanotroph abundance at the upstream and downstream sites under field conditions, respectively (Figure 5). Through the amended ^13^CH_4_ concentrations, *Methylobacter* increased in relative abundance by 14–29% in winter and 0–12% in summer. In addition, the *pmoA* sequencing of two peak fractions in the Bali mangrove soils yielded similar methanotrophic compositions.

In addition, the Type II methanotroph *Methylocystis* accounted for 10–15% of the relative abundance in the fresh soils from the upstream site but less than 5% in the fresh soils from the downstream site in both winter and summer under field conditions. Moreover, the relative abundance of *Methylocystis* in ^13^C-DNA decreased after the incubation experiment with ^13^CH_4_ in both mangrove soils in winter. However, the abundance remained almost constant in both mangrove soils in summer.

The results from high-throughput sequencing of the 16S rRNA genes revealed very different community structures. Overall, 2–20% of the methanotrophic populations were identified in the ^13^CH_4_-amended heavy fractions based on the 16S rRNA sequencing. Although Type Ia methanotrophs were still mostly dominant in both mangrove soils, 16S rRNA gene sequencing revealed that *Methylobacter* and *Methylomonas* were the major Type Ia methanotrophs in the collected fresh soils (*p* < 0.05), while *pmoA* gene sequencing suggested that it was just *Methylobacter* (Figure 6). Moreover, the two methanotrophs, *Methylobacter* and *Methylomonas*, were both active after being incubated in ^13^CH_4_-amended environments (*p* < 0.05). In addition, about 10–40% of the active methanotrophs belonged to the unclassified Methylococcaceae and were dominant in the fresh and incubated mangrove soils in both seasons (*p* < 0.05). Moreover, the 16S sequencing of the two peak fractions in the Bali mangrove soils showed slightly more *Methylomonas* and *Methyloprofundus* in fraction 7 than fraction 10, while both Type Ib and Type II methanotrophs had negligible relative abundances in both fractions.

The results from CCA showed the associations among environmental factors and different methanotrophic communities (Figure 7). Type I methanotrophs such as *Methylobacter* and *Methylomonas* were not influenced by the increasing salinity, while the Type II *Methylocystis* decreased with the increasing salinity.

## 4. Discussion

### 4.1. Composition and Adaptation of Methanotrophs in Field Mangrove Soils

This study demonstrates that different sections of an estuary can provide distinct ecological niches for methanotrophs in mangrove soils and influence methanotroph compositions. This may be because there is a wider variation in soil inundation statuses at the downstream than the upstream site, making the methanotrophic bacteria more active and resulting in the downstream site having the most *pmoA* copies in winter. Moreover, because methanotrophs can use CH_4_ as their sole C source [39], the high TOC and S_b_OC in the upstream site soils appears to be independent of the CH_4_ oxidation potentials in the studied mangrove ecosystem.

In addition, the qPCR experiments showed that *pmoA* gene copy numbers were highest at the downstream site in winter, which is more evidence that there is ecological niche differentiation between the downstream and upstream sites (e.g., the downstream site has a more dynamic soil aeration status). Furthermore, the inconsistencies in methanotrophic activities and *pmoA* gene increments may indicate that methanotrophic community composition is more important than abundance when determining the CH_4_ oxidation rates in mangrove soils. In addition, the heavy fraction enriched with ^13^C-*pmoA* genes in the upstream in summer was found at the 10th layer, which may indicate that little ^13^C was assimilated into the microbial genes and instead may have just respired as ^13^CO_2_, and thus many of the *pmoA* genes in fraction 10 may not been ^13^C labeled. This assumption may be confirmed by the two sequenced fractions (i.e., 7th and 10th fractions) in the Bali mangrove soils, because both fractions showed similar relative methanotrophic abundances. Moreover, the low percentages (i.e., 2%) of methanotrophic populations in the overall microbial community based on the 16S rRNA sequencing in the Guandu soils in summer may also confirm that few methanotrophs were ^13^C labeled.

The methanotrophic community compositions, measured using high-throughput sequencing, further revealed a potential reason for the high CH_4_ oxidation potentials in the downstream mangrove soils. Although the results showed that Type I methanotrophs dominate both mangrove soils, the Type II methanotrophs decreased in relative abundance from 10–20% upstream to 3–6% downstream (closer to the estuary of the Tamsui River). Type II methanotrophs are reported to have lower CH_4_ oxidation potentials than Type I methanotrophs in environments where the CH_4_ concentration frequently fluctuates [20,21], so the higher Type I methanotrophs in the downstream soils may be a result of heterogeneous CH_4_ oxidation at microsites under inundation and salinity stresses due to daily tides.

There were low counts of Type Ib methanotrophs belonging to the OSC-rel group and the Type II methanotroph *Methylocystis*, and high counts of Type Ia *Methylobacter* in the studied mangrove forests, suggesting that soil salinity regulates the distribution and growth of methanotrophic bacteria. Previous studies have indicated that Type Ib and Type II methanotrophs have low salinity resistance [23,40], while Type Ia methanotrophs are commonly found in saline and hypersaline environments [18,19,22,23]. Thus, our current study demonstrates that the changes in ecological niches can influence the compositions of methanotrophic communities at a spatial scale from upstream to downstream. Furthermore, the Type I methanotrophs levels were positively correlated with soil labile nutrient concentrations (i.e., TDN, NH_4_, and S_b_ON) from CCA, supporting the theory that they are r-selected organisms [9]. However, the observed methanotrophic compositions from our present study showed that the relative abundance of *Methylobacter* did not change much across the river sites. Instead, other methane-oxidizing bacteria that are yet to be classified (i.e., methane-oxidizing bacteria like (MOB-like)) appeared to be more abundant at the downstream site. This implies that more undiscovered salt-tolerant methanotrophs exist in coastal and marine ecosystems.

In addition, Type Ia *Methylobacter* was the dominant methanotrophic bacterium in both upstream and downstream sites. However, results from this study and our previous study located between the two mangrove sites in the present study [19] show that there is no transitional change in compositions. In our previous study sites, Type Ia *Methylosarcina* and uncultured Type Ib methanotrophs in the deep-sea-5 cluster were the dominant genera and shared >50% of the relative abundance. Moreover, one of the sites contained nearly 20% Type II methanotrophs, including *Mehtylocystis*, *Methylosinus,* and *Methylosinus/cystis*. Although the exact mechanism for this inconsistency in methanotrophic compositions is still unclear, we noticed that mangroves in the previous study grew on something resembling a deposition bar, which was only inundated during monthly high tides, and thus had low salinity. This difference in soil salinity along the Tamsui estuary may cause niche differentiation in the mangrove soils and alter methanotrophic community compositions. This assumption was supported by the differences in soil redox potentials: 180–370 mV in our previous study sites and about −200 mV in the present study sites.

Climate change and warming global temperatures appear to increase litter decomposition in mangrove forests [41], which likely accelerates the C cycle in and increase greenhouse gas emissions from mangrove soils [42]. On the other hand, the global warming-induced salt water intrusions in coastal ecosystems may also increase the salinity by 1–3 psu [43], which likely increases the Type Ia methanotrophic population and methane uptake capacity. Moreover, the competitions between sulfate reducing bacteria and methanogens for C sources may also shift under increased SO_4_^2-^ due to global warming [42,44]. All of these factors likely reduce the overall CH_4_ flux and increase CO_2_ flux from mangrove ecosystems. However, more studies are needed to confirm these above-mentioned hypotheses.

### 4.2. Active Methanotrophs Identified with ^13^CH_4_ DNA-SIP

High-throughput sequencing of ^13^CH_4_-labeled *pmoA* heavy fraction DNA revealed that Type Ia *Methylobacter* is the most active methanotroph in the mangrove forest soils in both winter and summer. In addition, the CH_4_ oxidation potential at the downstream site in winter was found to be significantly higher than that in summer. However, the abundance of methanotrophic bacteria (Figure 3) and soil chemical properties (Table 1) remained largely unchanged between the two sampling seasons. Moreover, both methanotrophic communities were dominated by *Methylobacter* (>50%). Although we did not measure the soil temperature during the field soil sampling, the air and water temperature typically vary 10–15 °C in the Tamsui estuary between winter (i.e., 18.3–18.6 °C) and summer (i.e., 30.5–31.5 °C). Thus, it is plausible that *Methylobacter* is more active at low temperatures. A previous study showed that *Methylobacter psychrophilus* grows optimally at 6–10 °C in a pure culture laboratory [45]. Furthermore, previous studies observed an uncultured *Methylobacter* sp. from the cold Zoigê Tibetan Plateau wetland in China that was adapted to environmental temperatures of 4–21 °C [18,46]. The *Methylobacter* identified from this research was distantly related to the *Methylobacter* sp. found in the Zoigê wetland (Appendix A), and this may strengthen our assumption that the observed high CH_4_ oxidation potential is associated with high *Methylobacter* abundance in the Bali mangrove forests in winter.

However, the *Methylobacter* genes identified in our previously studied mangrove forest between the two sites in the present study was more closely related to *Methylomicrobium*-jap-rel genes than the *Methylobacter* ones in this study. This result implies that species in *Methylobacter* genera may be adapted to different soil salinities and temperature ranges. Moreover, *Methylomicrobium* was categorized as a sub-branch of *Methylobacter* and was divided into an adjacent subbranch until 1995 [47], so the *Methylobacter* and *Methylomicrobium*-like methanotrophs found in our previous study may be less active in polyhaline environments.

In addition, the ~10% of unclassified Type Ia methanotrophs at the downstream site increased to a relative abundance of 20% after amended CH_4_ incubation. Because a similar observation was reported in the tidal-influenced mangrove soil in our previous study [19], the unclassified methanotrophs in this study may also be new methanotrophic bacteria adapted to highly saline environments.

One notable point is that, because we only sequenced DNA in the fractions with the most abundant *pmoA* gene copies, we might have overlooked some Type Ia methanotrophs with more T and A nucleotides and some Type Ib and Type II methanotrophs with more C and G nucleotides.

### 4.3. pmoA and 16S rRNA Sequences

The high-throughput sequencing of 16S rRNA genes resulted in a considerable number of unclassified Methylococcaceae after amended ^13^CH_4_ incubation. This observation was similar to our previous finding in another mangrove forest [19] in which unclassified Methylococcaceae made up more than 50% of the sequences at all sites. The mangrove forests in this present study were spatially connected to the previously-analyzed mangrove forests, so the unclassified methanotrophs in the upstream and downstream sites may also be novel methanotrophs that are important to CH_4_ oxidation in the mangrove forest ecosystems [19].

In addition, the Type Ib *Methyloparacoccus* was identified only from 16S rRNA gene sequencing but not from the *pmoA* gene because the database we used for classification did not include the *pmoA* gene fraction of *Methyloparacoccus* [48]. However, the phylogenetic tree of methanotrophic communities using *pmoA* sequencing showed some unclassified Type Ib methanotrophs in the study sites that were distantly related to *Methyloparacoccus* (Appendix A), implying that the unclassified Type Ib methanotrophs are likely *Methyloparacoccus* in the present studied sites.

The ^13^C labeled 16S rRNA gene analysis revealed a considerable number of Type Ia *Methylomonas* that were active at the downstream site. The reason for the high abundance of this phylotype remains unclear, but it highlights the importance of cultivating novel methanotrophs using state-of-the-art techniques such as the single-cell method. Furthermore, phylogenetic congruency between *pmoA* and 16S rRNA genes warrant further study to better represent and track the population dynamics of methanotrophs in natural settings [49]. Indeed, the phylogenetic tree of methanotrophic communities using 16S rRNA sequencing found that *Methylobacter*, *Methylomonas,* and *Methylosarcina* identified in this study are distantly related (Appendix A). This may explain the inconsistencies between the methanotrophic communities identified with *pmoA* and 16s rRNA genes. Further studies may consider sequencing the entire 16S rRNA (V1–V9) in the mangrove soils to identify members of the methanotrophic community at more depth. Furthermore, the phylogenetic congruency between *pmoA* and 16S rRNA genes warrants further study to better represent and track the population dynamics of methanotrophs in natural settings [49].

## 5. Conclusions

Our study indicates that methanotrophic species and abundance can vary across space in mangrove forests along a river due to different niches. The high salinity in the downstream estuary may facilitate the growth and activity of Type Ia methanotrophs such as *Methylobacter* and *Methylomicrobium* and certain Type Ib methanotrophs yet to be classified. On the other hand, Type II methanotrophs did not grow well in polyhaline environments. Moreover, the high salinity-tolerant *Methylobacter* identified in the mangrove forests may also be more active in low temperature soil environments. Overall, this study demonstrates that predominant methanotrophs vary widely depending on not only the type of freshwater ecosystem they live in but also fine-scale soil physiochemical properties.

## Figures and Tables

**Figure 1 microorganisms-08-01248-f001:**
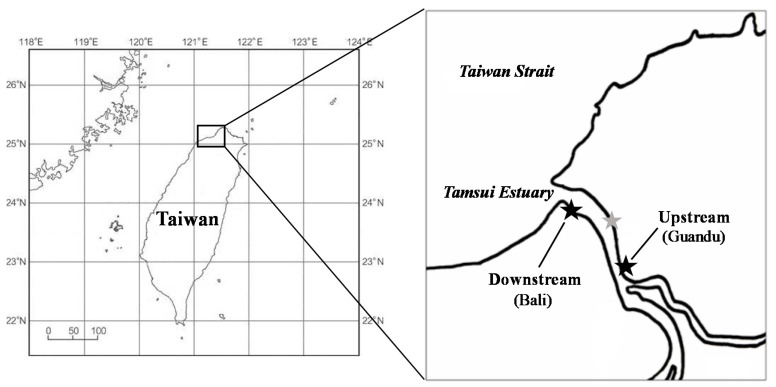
Map of studied mangrove sites (black stars) in the Tamsui estuary, Taiwan. The grey star indicates the site from a previous study [19].

**Figure 2 microorganisms-08-01248-f002:**
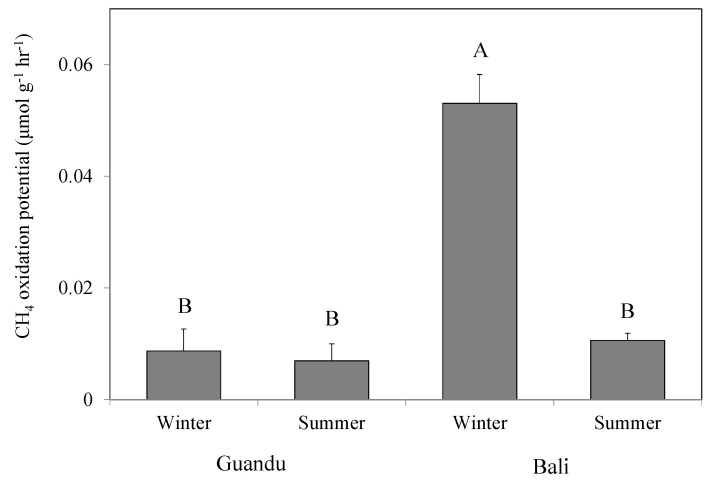
CH_4_ oxidation potential in the studied upstream (Guandu) and downstream (Bali) mangrove soils. Bars with the same letters are not significantly different at *p* = 0.05 based on Tukey’s honestly significant difference (HSD) comparison.

**Figure 3 microorganisms-08-01248-f003:**
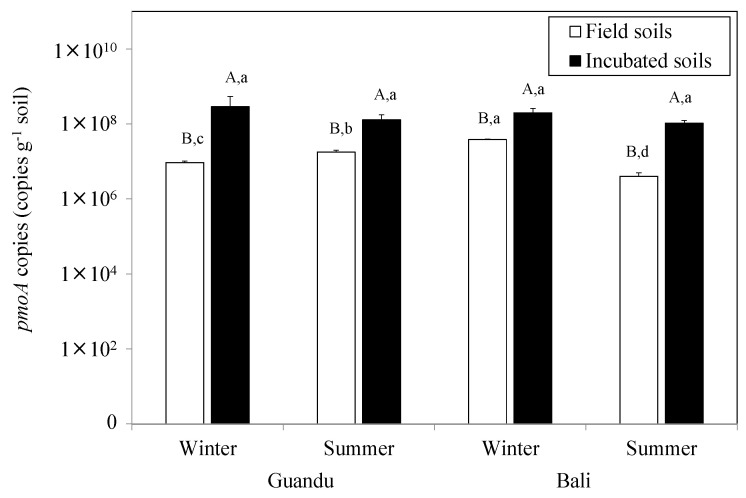
Number of *pmoA* copies in the studied upstream (Guandu) and downstream (Bali) mangrove soils. Bars with the same letters are not significantly different at *p* = 0.05 based on Tukey’s HSD comparison. The capital letters indicate the statistical results from *pmoA* genes before and after incubation at one site, and the lower case letters indicate the statistical results from *pmoA* genes between the sites and seasons in the fresh or incubated soils.

**Figure 4 microorganisms-08-01248-f004:**
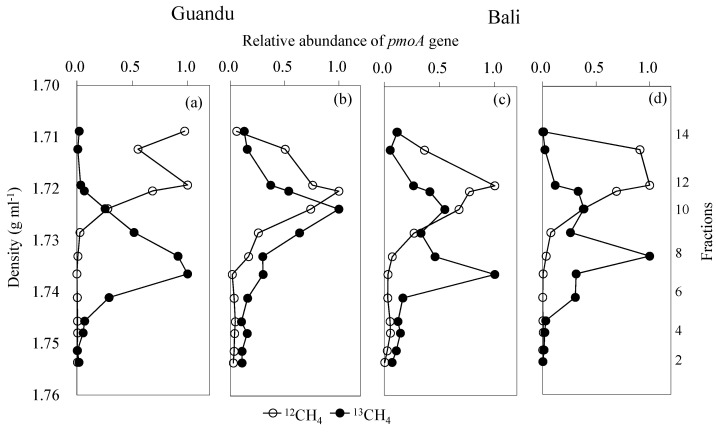
Relative abundance of the *pmoA* gene from ^12^CH_4_- and ^13^CH_4_-incubated soils in upstream (Guandu) and downstream (Bali) mangrove forests in winter (**a**,**c**) and in summer (**b**,**d**) at different density fractions (fractions 2–14).

**Figure 5 microorganisms-08-01248-f005:**
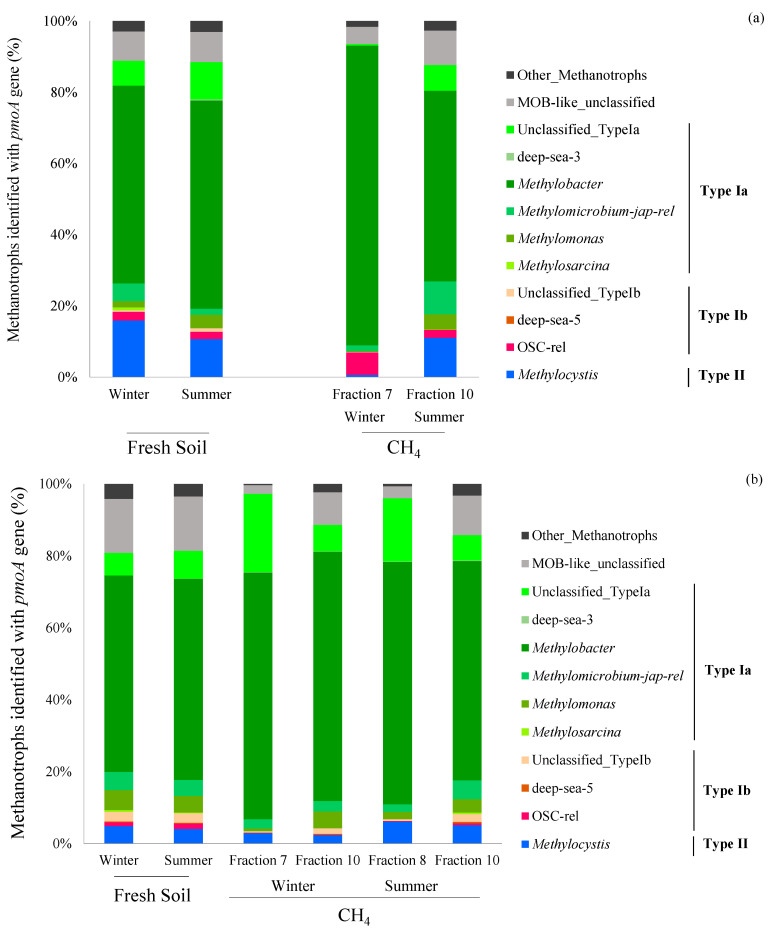
The relative abundance of methanotrophic communities identified with the *pmoA* genes in the average of triplicate fresh soils under field conditions and in ^13^C-DNA from ^13^CH_4_-enriched microcosms of (**a**) upstream (Guandu) and (**b**) downstream (Bali) mangrove forest soils, Taipei, Taiwan. Two regional peak fractions of *pmoA* genes were found in the ^13^CH_4_-amended Bali mangrove soils in one season (Figure 4), so the *pmoA* genes in both fractions were sequenced to identify the potential active methanotrophs.

**Figure 6 microorganisms-08-01248-f006:**
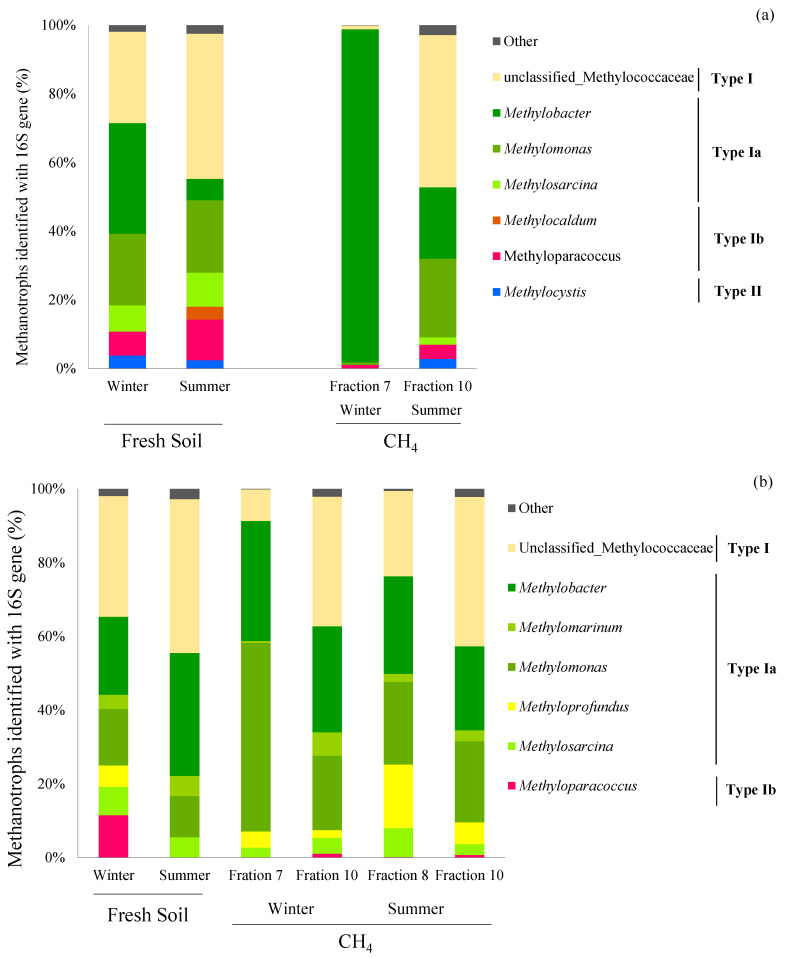
The relative abundance of methanotrophic communities identified with the 16S rRNA genes in fresh and ^13^CH_4_-enriched (**a**) upstream (Guandu) and (**b**) downstream (Bali) mangrove forest soils, Taipei, Taiwan. Two regional peak fractions of *pmoA* genes were found in the ^13^CH_4_-amended Bali mangrove soils in one season (Figure 4), so the 16S rRNA in both fractions were sequenced to identify the potential active methanotrophs.

**Figure 7 microorganisms-08-01248-f007:**
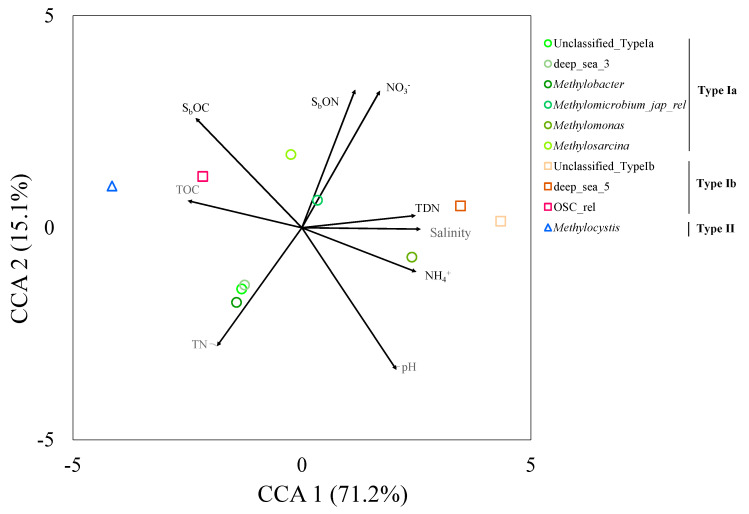
Canonical correspondence analysis (CCA) between methanotrophs and physiochemical properties in mangrove forests in Tamsui, Taipei, Taiwan. (S_b_OC: soluble organic C; S_b_ON: soluble organic N; NH_4_^+^: ammonium; NO_3_^-^: nitrate; TDN: total dissolved N; PMN: potential mineralizable N; TOC: total organic C; TN: total N.).

**Table 1 microorganisms-08-01248-t001:** Soil physiochemical properties of the studied mangrove soils.

Site	Season	Surface Water Salinity (psu)	pH	Soil Redox (mV)	Soil Texture	TOC	TN	S_b_OC	NH_4_^+^	NO_3_^−^	S_b_ON	TDN	PMN
						(%)		(mg C kg^−1^ Soil)	(mg N kg^−1^ Soil)				
Upstream (Guandu)	Winter	20	3.6 ± 0.0	−208 ± 73	silty clay	4.0 ± 0.8	0.3 ± 0.0	21.4 ± 1.8	10.0 ± 1.2	0.3 ± 0.1	21.0 ± 9.0	31.3 ± 7.9	14.3 ± 1.3
	Summer	17	5.7 ± 0.0			3.4 ± 0.0	0.3 ± 0.0	12.7 ± 2.5	7.7 ± 0.2	0.3 ± 0.0	12.8 ± 4.8	20.7 ± 4.9	11.9 ± 1.0
Downstream (Bali)	Winter	20	5.6 ± 0.0	−196 ± 16	clay loam	2.0 ± 0.1	0.3 ± 0.0	9.6 ± 2.0	6.5 ± 1.7	0.4 ± 0.1	18.3 ± 5.0	25.1 ± 6.2	32.6 ± 3.1
	Summer	21	6.2 ± 0.0			2.9 ± 0.0	0.3 ± 0.0	6.4 ± 3.8	54.2 ± 0.7	0.3 ± 0.0	27.7 ± 15.3	82.2 ± 15.4	20.7 ± 3.7

S_b_OC: soluble organic C; S_b_ON: soluble organic N; TDN: total dissolved N; PMN: potential mineralizable N.

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
