# Peer review of "Niche Differentiation of Active Methane-Oxidizing Bacteria in Estuarine Mangrove Forest Soils in Taiwan"

_microorganisms, 2020, doi:10.3390/microorganisms8081248_

Round 1
Reviewer 1 Report
It is suitable for publication in Microorganisms.
Reviewer 2 Report
The revised manuscript has been much improved, however, there are several issues to be addressed prior to publication in Microorganisms.
- L184: "These two soil properties were lower at….." This sentence should be rephrased because the TN values were all 0.3.
- L210-214: Fraction nos. should be indicated together with density. It is difficult to understand the correspondence between fraction nos. and plots in Figure 4. The range of density of concentrated fractions seems to be ca. 1.72-1.75. The peak of the heavy fraction in (b) seems to be at 1.725 (9th fraction?). What are the regional peak fractions? Which 10 fractions?
- L218: Should be "and in summer".
- In Figures 5 and 6, the relative abundance of methanotrophs from indicated fractions was presented for 13CH4-enriched samples of the downstream (Bali) mangrove forest. It should be mentioned why these fractions were presented. For the upstream sample, which fraction was used for the analysis? More detail information of samples used should be added to the legends to Figures 5 and 6.
Reviewer 3 Report
This is an interesting study and I do like the scientific question behind this paper and the approach the authors applied. It is not highly novel, but it does add insight on the topic. The environmental significance of Mangrove forest ecosystems in carbon cycling and their microbial ecology are timely research questions. Techniques and methods used are suitable and current. The research results and conclusions sound interesting but the manuscript itself would benefit from further clarification in parts (see below).
Comments:
Line 49: Replace “Pmo” with “pMMO”.
Lines 56-58: It would be helpful to add one or two references here for this sentence.
Figure 2: Could the bars in this figure please be filled with a solid colour, e.g. grey or black?
Figure 4: Error bars are missing in the graphs. The authors state that the “heavy” fraction for the 13CH4 Guandu summer gradient is fraction 10 (lines 211-213). However, this fraction is very much in the “light” region, in the same region as the main 12C peak of the 12C gradient. The “heavy” peak of this gradient would still be around fraction 6-7, it is not just as pronounced compared to the other gradients. Would it not be more probable to say that there was not a lot of 13C incorporation in this gradient/sample, compared to the other gradients? Fraction 10 will still contain a larger amount of unlabeled DNA, so will give very likely give mixed results for pmoA (and 16S) sequencing.
Figures 5 and 6: Why are the light and heavy SIP fractions shown for downstream/Bali (b) but not for upstream/Guandu (a)? The legend or text do not explain this. It would have been good to show either this or the respective 12CH4 light fractions.
Line 222-223: Should this maybe say something like “of bacterial methanotroph abundance”, instead of just “of bacterial abundance”?
What are the percentages of the methanotrophic population within the overall microbial community based on the 16S rRNA gene amplicon sequencing?
The CCA analysis, including Figure 7, and the related results description should be placed within the Results section.
Mangrove forests are under growing threat, e.g. from climate change. If possible, could the authors elaborate a bit what effect these environmental changes might have on the methanotrophic community and methane uptake capacity of these important ecosystems?
Author Response
see the attachment

This manuscript is a resubmission of an earlier submission. The following is a list of the peer review reports and author responses from that submission.
Round 1
Reviewer 1 Report
In this manuscript, the authors investigated methane oxidation and methanotroph community composition in upstream and downstream mangrove forest soils of an estuary in Taiwan. They revealed that methanotroph species and abundance changed seasonally and spatially depending on salinity. Overall, the experiments were well-designed and the authors presented solid data. This paper will provide an important contribution to understanding how to shape methanotroph communities in natural environment. However, there are several issues to be addressed prior to publication in Microorganisms.
Major points
1. The abstract does not reflect the conclusions and should be revised properly.
1-1. "salinity levels differ between winter and summer": According to Table 1, the seasonal change of salinity was not presented.
1-2. "methanotrophic populations undergo dynamic changes during aerobic methane oxidation": What does "during aerobic methane oxidation" mean? Which results did lead to this statement?
1-3. "Type Ia Methylobacter has a high tolerance to high salinity environments": This has been reported previously [19,20] and is not revealed in this study.
1-4. "several novel methanotrophs were also found in fresh and incubated soils at downstream site": Which strains (or sequence) in Figure S1 or S2 are novel methanotrophs? Unclassified Type I methanotrophs were also found in soils at upstream site (Fig. 6a).
1-5. "novel methanotrophs are present in mangrove forest soils": Novel methanotrophs are present in various natural environments.
1-6. "the global methane budget should be reassessed by …..": This is not discussed in the main text.
2. Methane oxidation activity and pmoA copy number were highest in winter at downstream site (Figs 2 and 3) and the authors speculated that the high salinity tolerant Methylobacter may be active in low temperature soil environments (L309-310). This finding is one of the interesting points of this study and should be much more discussed in the discussion section.
Minor points
- L28: "over 100 years" should be after "gas emissions".
- L42-43: "r-selected" and "K-selected" should be explained further.
- Table 1: "Surface water salinity" needs unit (psu).
- L182: According to Figure 4, the soil collected upstream in summer (B) had the heavy fraction of the pmoA genes located in the 10th layer (density: 1.72 g mL-1). Is the legend to Figure 4 correct?
- L206: Names of two methanotrophs should be written.
- Figure S2 is not referred in the main text.
Author Response
Major points
- The abstract does not reflect the conclusions and should be revised properly.
Response: We have thoroughly revised the abstract to reflect the conclusion of the study better.
1-1. "salinity levels differ between winter and summer": According to Table 1, the seasonal change of salinity was not presented.
Response: We have revised the salinity data in Table 1 to reflect the seasonal variations.
1-2. "methanotrophic populations undergo dynamic changes during aerobic methane oxidation": What does "during aerobic methane oxidation" mean? Which results did lead to this statement?
Response: We removed the sentences to avoid confusions.
1-3. "Type Ia Methylobacter has a high tolerance to high salinity environments": This has been reported previously [19,20] and is not revealed in this study.
Response: We followed the comments and revised the sentence to read “Methylobacter and Methylomicrobium-like Type I methanotrophs dominated methane-oxidizing communities in the field conditions and were significantly 13C-labeled in both upstream and downstream sites, while Type Ia Methylobacter were well adapted to high salinity and low temperature” to avoid confusion. (Lines 19-22)
1-4. "several novel methanotrophs were also found in fresh and incubated soils at downstream site": Which strains (or sequence) in Figure S1 or S2 are novel methanotrophs? Unclassified Type I methanotrophs were also found in soils at upstream site (Fig. 6a).
Response: We have updated the figures and made sure that the unclassified Type I methanotrophs are now in both figures.
1-5. "novel methanotrophs are present in mangrove forest soils": Novel methanotrophs are present in various natural environments.
Response: We revised the sentence to read “These results suggest that Type I methanotrophs displays niche differentiation associated with environmental divergence between upstream and downstream mangrove soils, and this highlights its importance for cultivating novel methane oxidizers in mangrove forest soils” to avoid confusions. Thank you for the comments. (Lines 27-29)
1-6. "the global methane budget should be reassessed by …..": This is not discussed in the main text.
Response: Thank you very much for the comments. We have removed the sentences to avoid confusions.
- Methane oxidation activity and pmoA copy number were highest in winter at downstream site (Figs 2 and 3) and the authors speculated that the high salinity tolerant Methylobacter may be active in low temperature soil environments (L309-310). This finding is one of the interesting points of this study and should be much more discussed in the discussion section.
Response: Thank you very much for the suggestion. We add a paragraph in 4.2 Section to emphasis the potential influence of temperature to Methylobacter to read “In addition, the methane-oxidizing potential at the downstream site in winter was found to be significantly higher than that in summer. However, the abundance of methanotrophic bacteria (Figure 3) and soil chemical properties (Table 1) remained largely unchanged between the two sampling seasons. Moreover, both methanotrophic communities were dominated by Methylobacter (>50%). Although we did not measure the soil temperature during the field soil sampling, the air and water temperature typically vary 10–15°C in the Tamsui estuary between winter and summer. Thus, it is plausible that Methylobacter is more active at low temperatures than high ones”. (Lines 315-321)
Minor points
L28: "over 100 years" should be after "gas emissions".
Response: We are sincerely sorry about the confusion. The “100 years” in our original draft meant the time horizon of CH4 comparing CO2 in trapping heat. We have revised the sentence to read “Global warming is caused by anthropogenic greenhouse gas emissions, and the major greenhouse gas, methane (CH4), traps 25 times more heat than carbon dioxide (CO2) over a 100-year time horizon” to avoid the confusion. (Line 36)
L42-43: "r-selected" and "K-selected" should be explained further.
Response: We followed the suggestion and revised the sentences to read “Type I methanotrophs are often considered r-selected species because they have high reproduction rates but low survival rates under adverse environments. On the other hand, Type II methanotrophs are adapted to oligotrophic and fluctuating environments, but have low growth rates under changing environments, and are thus considered K-selected organisms (i.e. low reproduction rates but high survival rates)” to hopefully better explain the ecological strategies. (Lines 59-63)
Table 1: "Surface water salinity" needs unit (psu).
Response: We followed the suggestion and added “psu” in Table 1.
L182: According to Figure 4, the soil collected upstream in summer (B) had the heavy fraction of the pmoA genes located in the 10th layer (density: 1.72 g mL-1). Is the legend to Figure 4 correct?
Response: Thank you very much for pointing out the error we made. We have revised the sentence to read “Gradient fractionation of the 13CH4-incubated soil DNA showed that the 13C-labeled pmoA genes were mostly concentrated in the 7–8 fractions (density: 1.727–1.732 g mL-1), except in the soil collected upstream in summer, in which the heavy fraction of the pmoA genes was located in the 10th layer (density: 1.72 g mL-1) (Figure 4)”. (Lines 204-207)
L206: Names of two methanotrophs should be written.
Response: We followed the suggestion and revised the sentence to read “Moreover, the two methanotrophs, Methylobacter and Methylomonas, were both active after being incubated in 13CH4-amended environments”. (Lines 237-238)
Figure S2 is not referred in the main text.
Response: Thank you very much for pointing out the error. We have added the link of Figure S2 in the main text in Line 353.
Reviewer 2 Report
Authors investigated the methane oxidation activities and community structures of methanotrophic bacteria in soils at two sites in an estuarine mangrove forest. They compared the data observed and tried to reveal ecological niche of methanotrophic bacteria in such coastal area. Result section concisely explains outcome from the experiments, and it could be a concern of the readers of Microorganisms; overall, current manuscript lacks clarity and has many serious flaws. This makes authors’ conclusions very ambiguous. This manuscript should be revised extensively prior to consideration of publication.
Major comments:
1. L 169-172. Authors stated that the methane oxidation potential (Fig. 2) was reflected in their methanotrophic bacterial communities. From the data of this study, it is unclear whether these are correlated. The high methane oxidation activity seen in winter Bali cannot be explained by amount of pmoA copies before or after incubation (Figure 3). The gene copy number in another site or season also is high enough, but the activities are low. Please reconsider these phrases.
2. L 181-183. Authors stated exception of the case ‘downstream in winter’. From the density value of 1.72 g/mL in Fig. 4, ‘upstream in summer’ looks correct. If so, how do authors interpret this result in Discussion section?
3. L 226-233. Authors stated a possible reason for the high metabolic activity in winter Bali. Although they pointed the decrease in abundance of Type II bacteria; but it decreased similarly in summer Bali, wherein the activity was still low. So, it is unclear that abundance of Type II bacteria accounts for the methane oxidation activity. Reconsider these phrases.
4. Authors tried to explain the distribution of pmoA gene, likely related with abundance of methanotrophs, and the methanotroph compositions in the two sites. As described above, however, these experimental data appear not to be interpreted objectively. Authors should overall rearrange the Discussion section, followed by reconsideration of Conclusion and Abstract sections.
Minor comments:
5. L 35-39. Please explain the classification system (type) in methanotrophic bacteria.
6. L 45. Please explain the function of pmoA gene.
7. L 243-244, Fig. 7. How about Type Ib methanotrophs? One appears to decrease with increasing salinity, while two increase with increasing salinity.
8. L 258-260. Please describe the previous results concretely (ref. 16) and clearly state what is outcome from their two studies.
Author Response
- L 169-172. Authors stated that the methane oxidation potential (Fig. 2) was reflected in their methanotrophic bacterial communities. From the data of this study, it is unclear whether these are correlated. The high methane oxidation activity seen in winter Bali cannot be explained by amount of pmoA copies before or after incubation (Figure 3). The gene copy number in another site or season also is high enough, but the activities are low. Please reconsider these phrases.
Response: We followed the suggestion and added sentences to read “Furthermore, the inconsistencies in methanotrophic activities and pmoA gene increments may indicate that methanotrophic community composition is more important than abundance when determining the methane oxidation rates in mangrove soils” to explain the potential reason of this inconsistence. (Lines 258-260)
- L 181-183. Authors stated exception of the case ‘downstream in winter’. From the density value of 1.72 g/mL in Fig. 4, ‘upstream in summer’ looks correct. If so, how do authors interpret this result in Discussion section?
Response: Thank you very much for pointing out the error we made. We have revised the sentence as suggestion and added explanation to read “In addition, the heavy fraction enriched with 13C-pmoA genes in the upstream in summer was found at the 10th layer, which indicates that little 13C was assimilated into the microbial genes, and instead may have just respired as 13CO2”. (Lines 260-263)
- L 226-233. Authors stated a possible reason for the high metabolic activity in winter Bali. Although they pointed the decrease in abundance of Type II bacteria; but it decreased similarly in summer Bali, wherein the activity was still low. So, it is unclear that abundance of Type II bacteria accounts for the methane oxidation activity. Reconsider these phrases.
Response: Thank you very much for your comments. We added a paragraph to explain possible reasons that caused the different methane oxidation rates in the downstream site in winter and summer. The explanations are as follow: In addition, the methane-oxidizing potential at the downstream site in winter was found to be significantly higher than that in summer. However, the abundance of methanotrophic bacteria (Figure 3) and soil chemical properties (Table 1) remained largely unchanged between the two sampling seasons. Moreover, both methanotrophic communities were dominated by Methylobacter (>50%). Although we did not measure the soil temperature during the field soil sampling, the air and water temperature typically vary 10–15°C in the Tamsui estuary between winter and summer. Thus, it is plausible that Methylobacter is more active at low temperatures than high ones. (Lines 351-321)
- Authors tried to explain the distribution of pmoA gene, likely related with abundance of methanotrophs, and the methanotroph compositions in the two sites. As described above, however, these experimental data appear not to be interpreted objectively. Authors should overall rearrange the Discussion section, followed by reconsideration of Conclusion and Abstract sections.
Response: Thank you very much for your suggestion. We have revised the Abstract, Discussions and Conclusion to hopefully better explain the results of this study.
Minor comments:
- L 35-39. Please explain the classification system (type) in methanotrophic bacteria.
Response: We followed the suggestion and added the explanations about the classification system in methanotrophic bacteria to read “Methanotrophic communities can generally be categorized into three types: Type I, which includes family Methylococcaceae (γ-Proteobacteria) [13]; Type II, which includes families Methylocystaceae and Beijerinckiaceae (α-Proteobacteria); and Type X, which includes some other γ-Proteobacteria”. (Lines 46-49)
- L 45. Please explain the function of pmoA gene.
Response: We followed the suggestion and added the explanations about the pmoA gene to read “Almost all of these above mentioned methanotrophs possess the particulate methane monooxygenase (Pmo), which catalyzes methane oxidation”. (Lines 49-50)
- L 243-244, Fig. 7. How about Type Ib methanotrophs? One appears to decrease with increasing salinity, while two increase with increasing salinity.
Response: We followed the suggestion and added explanation in the 4.1 Section to read “It is noteworthy that some methanotrophs in the unclassified Type Ib and deep-see-5 cluster also appeared to resist saline stress, implying that some certain γ-Proteobacteria methanotrophs may also be adapted to high salinity”. (Lines 280-282)
- L 258-260. Please describe the previous results concretely (ref. 16) and clearly state what is outcome from their two studies.
Response: We followed the suggestion and add sentences to state the outcomes from our previous study to read “In our previous study sites, Type Ia Methylosarcina and uncultured Type Ib methanotrophs in the deep-sea-5 cluster were the dominant genera and shared >50% of the relative abundance. Moreover, one of the sites contained nearly 20% of the Type II methanotrophs, including Mehtylocystis, Methylosinus, and Methylosinus/cystis”. (Lines 303-306)
Round 2
Reviewer 2 Report
Authors have revised thoroughly according to the reviewer's comments. The current manuscript is acceptable for readers of Microorganisms.